# Food Contaminants Effects on an In Vitro Model of Human Intestinal Epithelium

**DOI:** 10.3390/toxics9060135

**Published:** 2021-06-09

**Authors:** Marion Guibourdenche, Johanna Haug, Noëllie Chevalier, Madeleine Spatz, Nicolas Barbezier, Jérôme Gay-Quéheillard, Pauline M. Anton

**Affiliations:** 1PériTox—Périnatalité & Risques Toxiques, UMR-I 01 INERIS, Université Picardie Jules Verne, 80025 Amiens, France; marion.guibourdenche@outlook.fr (M.G.); jerome.gay@u-picardie.fr (J.G.-Q.); 2Institut Polytechnique UniLaSalle, Université d’Artois, ULR 7519, 19 rue Pierre Waguet, BP 30313, 60026 Beauvais, France; johanna.haug@orange.fr (J.H.); noellie.chevalier_pc@yahoo.fr (N.C.); madeleine.spatz@inrae.fr (M.S.); nicolas.barbezier@unilasalle.fr (N.B.)

**Keywords:** intestinal barrier, Caco-2/TC7, HT29-MTX, chlorpyrifos, AGEs, food contaminants

## Abstract

Pesticide residues represent an important category of food contaminants. Furthermore, during food processing, some advanced glycation end-products resulting from the Maillard reaction can be formed. They may have adverse health effects, in particular on the digestive tract function, alone and combined. We sought to validate an in vitro model of the human intestinal barrier to mimic the effects of these food contaminants on the epithelium. A co-culture of Caco-2/TC7 cells and HT29-MTX was stimulated for 6 h with chlorpyrifos (300 μM), acrylamide (5 mM), N^ε^-Carboxymethyllysine (300 μM) alone or in cocktail with a mix of pro-inflammatory cytokines. The effects of those contaminants on the integrity of the gut barrier and the inflammatory response were analyzed. Since the co-culture responded to inflammatory stimulation, we investigated whether this model could be used to evaluate the effects of food contaminants on the human intestinal epithelium. CPF alone affected tight junctions’ gene expression, without inducing any inflammation or alteration of intestinal permeability. CML and acrylamide decreased mucins gene expression in the intestinal mucosa, but did not affect paracellular intestinal permeability. CML exposure activated the gene expression of MAPK pathways. The co-culture response was stable over time. This cocktail of food contaminants may thus alter the gut barrier function.

## 1. Introduction

For years now, agricultural practice and the increased consumption of processed foods have exposed humans to a larger range of food contaminants which threaten the health and safety of consumers [1]. Indeed, on top of pesticides residues, which are among the most common food contaminants, food processing, and particularly heat treatment, is at the origin of a non-enzymatic glycation reaction also called the Maillard reaction. This reaction generates a large collection of compounds contributing to the odor, flavor and color of the food matrix [2], but also many others that may be considered life threatening. One of the main challenges is to understand the consequences of the exposure to many food contaminants for our health and, more significantly, to evaluate their impact on the digestive tract, the main route of exposure to food contaminants. The use of in vitro models to evaluate such effects is of prime importance to get objective information on their effect on our health.

Among pesticides, Chlorpyrifos (CPF) is an organophosphorus insecticide, still used worldwide in agriculture to protect fruits and vegetables crops and known to interfere with intestinal function. In a previous work, we showed that perinatal exposure to CPF alters the intestinal barrier function (tight junctions and mucus) [3,4,5,6]. Furthermore, CPF increases paracellular permeability in the human intestinal Caco-2 cell line, possibly through the transient modification of tight junction proteins’ structure [7]. As for the Maillard reaction, which generally occurs between reduced sugars and some amino acids [8], it leads to the production of several compounds among which are advanced glycation end products (AGEs). While their adverse effects on health are often suggested [9], it still remains difficult to confirm them [8]. For instance, the consumption of AGEs such as N^ε^-Carboxymethyllysine (CML) or of acrylamide has negative impacts on human health. Indeed, their increased presence in human tissues is correlated with oxidative stress, inflammation, and insulin resistance [10], and AGEs are involved in the development and progression of type 2 diabetes [11]. These food compounds may induce intestinal inflammation and alter the mucosal barrier and the intestinal permeability, but that has been sparsely documented, and little is known about the effects of concomitant exposure to the two families of food contaminants on the intestinal barrier and on the cellular mechanisms they may alter.

Models of in vitro human intestinal epithelium have been described in the literature in order to evaluate the toxicity and the bioavailability of substances [12,13] based on a mix of adequate cell lines. Caco-2 is an immortalized cell line of human colorectal adenocarcinoma commonly used to evaluate the modification of epithelial permeability following exposure to specific compounds. Those cells have the ability to differentiate spontaneously into enterocytes [14]. Nevertheless, since for many tumor cell lines their response tends to become heterogeneous after repeating passages [15], this has prompted scientists to generate clones such as TC7. This clone is known to better reproduce the functionality of intestinal cells in vivo with a more homogenous response over time [16]. Furthermore, to fully mimic the intestinal epithelial lining, there is a need for mixing enterocytes cells with goblet cells. The HT29-MTX cells are derived from HT29 cells fully differentiated into goblet cells [17]. As such, the co-culture of Caco-2 and HT29-MTX cells may constitute an interesting model to investigate intestinal epithelial functions [18,19] if the response remains stable with time and cell replication.

We aimed to use a Caco-2/TC7 and HT29-MTX co-culture model [20] to characterize, in a more homogeneous fashion, the effects of a cocktail of food contaminants (pesticides residues and AGEs) on the intestinal epithelium. We evaluated the putative effects of those molecules on gut permeability and barrier function (mucins and tight junction proteins expressions) and on two of the key intracellular signaling pathways involved in gut homeostasis: JAK/STAT and MAPKs [21,22,23].

## 2. Materials and Methods

### 2.1. Cell Culture Conditions

The Caco-2 cells/TC7clone expresses an intestinal epithelial phenotype when fully differentiated; it was used between passage 70 and 83. The human colorectal adenocarcinoma cell line HT29-MTX (European Collection of Authenticated Cell Cultures) was used between passage 85 and 93. This cell line expresses an intestinal goblet cell phenotype when fully differentiated. Cells were grown alone as a mono-culture (Caco-2/TC7) or as a co-culture (Caco-2/TC7 and HT29-MTX) to mimic the intestinal epithelial lining. The two cell lines were routinely grown in an atmosphere of 5% carbon dioxide at 37 °C in DMEM GlutaMAX added with 10% (*v*/*v*) heat-inactivated fetal bovine serum (FBS), 1% (*v*/*v*) non-essential amino acids and 1% (*v*/*v*) penicillin/streptomycin (Fisher Scientific SAS, Illkirch, France). The medium was changed every other day. Cells were split below confluency using a trypsin solution (0.25% Trypsin EDTA—Thermofisher Scientific, Illkirch, France).

### 2.2. Pro-Inflammatory Cytokines Stimulation of the Caco2/TC7 and HT29/MTX Co-Culture

#### 2.2.1. Comparison of the Mono-Culture and the Co-Culture Sensitivity to Pro-Inflammatory Cytokines

Caco-2/TC7 (mono-culture) or Caco-2/TC7 and HT29-MTX (co-culture) were seeded on 6-well polycarbonate membrane cell culture inserts with HD 0.4 µm pores at a density of 4 × 10^5^ cells (Corning, Dutscher, Issy-les-Moulineaux, France). Culture medium was distributed on the insert (1 mL) and in the well (2 mL) supporting the insert. This set of experiments was realized in order to assess the modification of the response of the co-culture to the induction of an inflammatory response observed within the mono-culture. Indeed, Caco-2 cells are known to be able to produce inflammatory mediators in response to pro-inflammatory stimuli [24], and particularly the TC7 clone [25]. Cells were grown for 21 days to reach full differentiation. The culture medium was changed every other day. The day before the stimulation, cells were rinsed with phosphate buffered saline (PBS) and put into a serum-free medium. After 24 h in the depleted medium, cells were stimulated with a cocktail of pro-inflammatory cytokines—TNF-α (20 ng/mL), IL-1β (1 ng/mL) and IFN-ϒ (10 ng/mL) (BioTechne, Lille, France)—for 2, 6 or 24 h for the mono-culture and for 6 h for the co-culture, as this was the exposure timepoint that achieved the best stimulation of the mono-culture.

At the end of the stimulation, the insert medium (called below “apical medium”) and well medium (called below “basal medium”) were harvested and stored at −80 °C until further measurement of markers of intestinal inflammation and permeability. The cell layers were harvested in TriZOL and processed immediately.

##### Modulation by the Cocktail of Cytokines of the Expression of Genes Coding for Proteins of the Pro-Inflammatory and Pro-Apoptotic Pathways

To confirm the pro-inflammatory stimulation of the mono- and the co-culture by the cocktail of pro-inflammatory cytokines, we first measured the modification of the expression of genes involved: 1) in the pro-inflammatory signaling pathways—C-X-C motif chemokine ligand 8 (*CXCL8*) coding for IL-8, mitogen-activated protein kinase 14 (*MAPK14*) coding for p38-α, mitogen-activated protein kinase 8 (*MAPK8*) coding for JNK1, mitogen-activated protein kinase-kinase-kinase 1 (*MAP3K1*) coding for MEKK1, signal transducer and activator of transcription 3 (*STAT3*), nuclear factor-kappa B (*NF-κB*), tumor necrosis factors α (*TNF-α*), myeloid differentiation primary myeloid response 88 (*MyD88*); and 2) in the pro-apoptotic pathways—Caspase 3 and Caspase 9 (*CASP3* and *9*).

Total RNA was isolated from cells using TriZOL (Fisher Scientific SAS, Illkirch, France) and RNA concentration was measured spectrophotometrically (NanoDrop 2000, Thermo Scientific, Illkirch, France). cDNA was obtained from 1 µg RNA using the QuantiTect Reverse Transcription Kit following the manufacturer’s instructions (Qiagen, Courtaboeuf, France). Absorbance ratios at 260/280 nm and at 260/230 nm were measured by spectrophotometry to assess the purity of the DNA samples.

Primers and SYBR Green PCR master mix were respectively purchased from Eurofins Scientific France (Nantes, France) and Qiagen (Courtaboeuf, France) (Table 1). QPCRs were run on a StepOnePlus Real-Time PCR System (Applied Biosystem, Foster City, CA, USA) and the data were processed with the OneStepPlus software. Reactions were performed in duplicate. Levels of amplified cDNA were quantified using the 2^−ΔΔCT^ method (ΔΔCt = ΔCt_exposed_ − mean ΔCt_control_) compared to Glyceraldehyde 3 phosphate dehydrogenase (*GAPDH*) considered as the best housekeeping gene from two different genes tested (*GAPDH* vs. *PPIA*).

##### Chemokine Secretion Following a Cell Stimulation

The levels of IL-8 secretion in both the basal and apical media of mono- and co-cultures were measured after stimulation with the cocktail of cytokines. Measurement of IL-8 in the media harvested at the end of stimulation were performed using an ELISA kit according to the manufacturer’s instructions (Human IL-8/CXCL8 DuoSet ELISA, #DY208, BioTechne, Lille, France) [26].

### 2.3. Effects of Food Contaminants on the Co-Culture Model

The evaluation of the effects of contaminants in the co-culture of Caco-2/TC7 and HT29-MTX was realized as described previously. In a preliminary study, Caco-2/TC7 were stimulated with: (i) Chlorpyrifos-ethyl (CPF) (O. O-diethyl-O- (3.5.6-trichloro-2-pyridinyl) phosphorothioate solution (LGC Standards Laboratories, Molsheim, France) at concentrations of 100 and 300 μM in 1% methanol [7], (ii) vehicle alone, (iii) N^ε^(6)-Carboxymethyllysine (CML) in culture medium (PolyPeptide Laboratories France SAS, Strasbourg, France) at concentrations of 100 µM and 300 µM [27] or with iv) acrylamide in culture medium (Sigma-Aldrich, St. Quentin Fallavier, France) at concentrations of 1 and 5 mM [24]. Based on a preliminaryset of experiments, only cells grown on inserts and stimulated for 6 h with 5 mM of acrylamide, 300 μM of CML or 300 μΜ of CPF were further processed.

#### 2.3.1. Viability of Caco2/TC7 Mono-Culture after Exposure to Food Contaminants

The absence of cell death following the exposure to food contaminants was evaluated by comparing cell counts of viable Caco-2/TC7 cells (100 µL of cell suspension) unstimulated and stimulated with the contaminants using flow cytometry (MACS Quant Analyzer, Miltenyi Biotec, Köln, Germany).

#### 2.3.2. Assessment of the Alteration of the Barrier Function by the Food Contaminants

##### Effects of the Food Contaminants on the Paracellular Permeability

The alteration of paracellular epithelial permeability was assessed by measuring the flux of the Fluorescein isothiocyanate (FITC)-dextran 4 kDa (Sigma-Aldrich, St. Quentin Fallavier, France) from the apical to the basal medium. FITC-Dextran was put in the apical medium (333.3 µg/µL–10 µL/well) at the beginning of the stimulation [28]. The presence of FITC-Dextran was quantified in duplicates from 100 µL of the apical medium (1:50) and of the basal medium (1:1) at the end of the stimulation using a fluorimeter (SpectraMax M2, Molecular Devices, San Jose, CA, USA). The sample emission spectrum (528 nm) was recorded upon excitation (485 nm) using second-derivative spectroscopy to eliminate background noise and light scattering. The results are expressed as µg/µL.

##### Effects of the Food Contaminants on the Expression of Genes Coding for Tight Junctions and Mucins

We also evaluated the alteration of the epithelial lining by measuring the expression of genes coding for 1) tight junctions and microvilli proteins (Occludin (*OCLN*), Claudin 4 (*CLDN4*), Tight junction protein 1 (*TJP1*) and Villin 1 (*VIL1*)) and 2) mucins (mucin 3 (*MUC3*) and mucin 5AC (*MUC5AC*)). Briefly and as already stated above, the cDNA generated from extracted RNA was amplified according to the conditions described above. Samples were measured in duplicate by qPCR using primers described in Table 1, as already mentioned.

#### 2.3.3. Modulation of Pro-Inflammatory and Pro-Apoptotic Signaling Pathways by the Food Contaminants

The variation in the expression of genes involved in (1) pro-inflammatory signaling pathways *CXCL8*, *MAPK14*, *MAPK8*, *MAP3K1*, *STAT3*, *NF-κB*, *TNF-α* and *MyD88* and (2) pro-apoptotic pathways *CASP3* and *CASP9* was evaluated by qPCR as described above, and the secretion of the IL-8 cytokine by epithelial cells was quantified using the human IL-8/CXCL8 DuoSet ELISA in response to a 6 h CPF, CML or acrylamide stimulation. Measures were performed in duplicates.

### 2.4. Statistical Analyses

Data were expressed as the mean ± Standard Error of the Mean (SEM) and were analyzed using the GraphPad Prism software (GraphPad Prism version 8.4.3 for Windows, GraphPad Software, San Diego, CA, USA). Student’s t-test was performed for comparisons of mono-culture and co-culture sensitivity to pro-inflammatory cytokine stimulation. A parametric one-way ANOVA analysis of variance was performed and, when the difference was statistically significant, a Bonferroni post hoc test was then applied to analyze the effects of food contaminants on this in vitro model of human intestinal epithelium. The threshold for statistical significance was set to *p* < 0.05. 

## 3. Results

### 3.1. Pro-Inflammatory Cytokines Stimulation of the Mono- and the Co-Culture

#### 3.1.1. Comparison of the Mono and Co-Culture Sensitivity to Pro-Inflammatory Cytokines

The incubation for 2 and 6 h of the Caco-2/TC7 mono-culture with pro-inflammatory cytokines was associated with a significantly (*p* = 0.0021 and *p* < 0.0001, respectively) greater expression of the *NF-κB* gene (5-fold and 8-fold, respectively) in comparison to the non-stimulated cells (CTL) (Figure 1a). Furthermore, *CXCL8* mRNA expression after 2, 6 and 24 h of stimulation was also significantly (*p* = 0.0003, *p* < 0.0001 and *p* = 0.0005, respectively) higher (210-, 180- and 29-fold, respectively) (Figure 1b). IL-8 protein secretion was significantly (*p* = 0.0012, *p* < 0.0001 and *p* < 0.0001) greater in the apical medium after 2 h (1004 ng/mL ± 211 ng/mL vs. 63 ng/mL ± 8 ng/mL; 16-fold), 6 h (1713 ng/mL ± 41 ng/mL vs. 45 ng/mL ± 5 ng/mL in CTL; 38-fold) and 24 h (2468 ± 47 ng/mL vs. 152 ng/mL ± 17 ng/mL in CTL; 16-fold), and also in the basal medium (*p* = 0.0287 and *p* = 0.0014, respectively) after 2 h (434 ng/mL ± 159 ng/mL vs. 27 ng/mL ± 1 ng/mL in CTL; 12-fold) and 24 h (522 ng/mL ± 110 ng/mL vs. 43 ng/mL ± 3 ng/mL in CTL; 16-fold) (Figure 1c,d, respectively).

Incubation of the Caco-2/TC7 and HT29-MTX co-culture for 6 h with the pro-inflammatory cytokines resulted in a statistically significant overexpression (*p* < 0.0001 and 34-fold; *p* < 0.0001 and 4-fold, respectively) of *NF-κB* and *CXCL8* mRNA in comparison to the non-stimulated cells (CTL) (Figure 1e,f). Moreover, IL-8 protein secretion after 6 h of stimulation was also significantly (*p* < 0.0001 and *p* = 0.0003, respectively) greater in apical (421 pg/mL ± 42 pg/mL vs. 12 pg/mL ± 1 pg/mL in CTL; 35-fold) and basal (17 pg/mL ± 2 pg/mL vs. 4 pg/mL ± 0.8 pg/mL in CTL; 4.5-fold) media (Figure 1g,h, respectively).

#### 3.1.2. Modulation by the Cocktail of Cytokines of the Expression of Genes Coding for Proteins of the Pro-Inflammatory and Pro-Apoptotic Pathways

We did not observe any significant MAPK gene activation after a 6 h incubation of the Caco-2/TC7 mono-culture with pro-inflammatory cytokines. Only the mRNA expression of *CASP9* was significantly lowered (−0.2-fold, *p* = 0.0338) as compared to the non-stimulated cells (CTL) (Figure 2a). These results were also observed for the Caco-2/TC7 and HT29-MTX co-culture after a 6 h incubation with the cocktail of cytokines. Furthermore, the mRNA expression of *STAT3* was significantly downregulated (−0.7-fold; *p* < 0.0001), while mRNA expression of *CASP3* was significantly upregulated (3.2-fold *p* = 0.0305) after a 6 h incubation of the co-culture (Figure 2b).

### 3.2. Effects of the Food Contaminants on the Mono- and Co-Culture

#### 3.2.1. Viability of the Mono-Culture after Exposure to Food Contaminants

The Caco-2/TC7 mono-culture was incubated with the selected food contaminants to ascertain the level of cell death and to determine the optimal concentration to work with for the rest of the study. Based on these preliminary results, incubation with 300 μM of CPF and CML and with 5 mM of acrylamide induced sufficient activation of the cells (confirmed by IL-8 secretion) without generating any cell death. These concentrations were then used for the rest of the experiments. In addition, the absence of a per se effect of methanol, used as the vehicle for CPF stimulation, was confirmed.

#### 3.2.2. Assessment of the Alteration of the Barrier Function by the Food Contaminants

The effects of food contaminants were studied only in the Caco-2/TC7 and HT29-MTX co-culture, which is physiologically more relevant than the mono-culture.

##### Effects of Food Contaminants on the Epithelial Permeability

Paracellular intestinal permeability impairment was estimated via the amount of FITC-Dextran transiting from the apical to the basal medium [28]. After a 6 h stimulation with CML and/or CPF, we did not observe any significant diffusion (*p* > 0.05) of FITC-Dextran through the epithelial intestinal layer (Figure 3a,b, respectively).

The mRNA expression of *TJP1* and *VIL1* was significantly upregulated (17.0-fold; *p* = 0.0005 and 3.0-fold; *p* = 0.0111, respectively vs. CTL) after a 6 h stimulation with CPF, but significantly lowered after a 6 h co-exposure to CPF/CML (17.6-fold; *p* = 0.0008 and 3.3-fold; *p* = 0.0048, respectively vs. CPF) (Figure 3c).

##### Modulation of Pro-Inflammatory and Pro-Apoptotic Signaling Pathways by the Food Contaminants

The mRNA expression of *NF-κB* was not significantly (*p* > 0.05) modified by a 6 h incubation with either CPF alone or in combination with CML (Figure 4a). In contrast, a 6 h incubation with CML significantly upregulated (3.6-fold *p* = 0.010) *CXCL8* mRNA expression, which was not observed after the 6 h incubation with CPF/CML (Figure 4a).

Furthermore, following a 6 h incubation with CML and/or CPF, we did not observe any change (*p* > 0.05) in IL-8 protein secretion in the apical or the basal media (Figure 4b,c, respectively).

A 6 h incubation of the co-culture with CML resulted in a significantly greater mRNA expression of *MAPK8* (5.9-fold *p* = 0.0118 vs. CTL), but also of *CASP3* (3.5-fold; *p* < 0.0014 vs. CTL and 3.7-fold *p* < 0.0011 vs. CPF/CML) (Figure 4d).

#### 3.2.3. Effects of the Glycotoxins on the Caco2/TC7 and HT29/MTX Co-Culture

##### Effects of the Glycotoxins on the Intestinal Barrier Function

A 6 h incubation of the co-culture with the two glycotoxins, CML and acrylamide, was not associated with any significant (*p* > 0.05) change in FITC-Dextran concentration in the apical and the basal media (Figure 5a,b).

The 6 h incubation with CML (300 µM) resulted in a significantly lower expression of TJP1 mRNA (−1.8-fold; *p* < 0.0332 vs. acrylamide) and of VIL1 mRNA (1.7-fold; *p* < 0.0284 vs. acrylamide). This incubation also statistically significantly upregulated the expression of MUC3 mRNA (−1.9-fold *p* < 0.0299 vs. CML). Furthermore, following the 6 h incubation with CML or acrylamide, the expression of MUC5AC mRNA was significantly lowered (*p* < 0.0001 for both) by 0.6-fold and 0.4-fold, respectively (Figure 5c).

##### Effects of the Glycotoxins on the Expression of Key Genes Involved in the Pro-Inflammatory and Pro-Apoptotic Signaling Pathways

*NF-κB* mRNA expression in the co-culture was modified by neither CML nor acrylamide (*p* > 0.05 vs. CTL) (Figure 6a). However, a 6 h incubation with CML significantly upregulated *CXCL8* mRNA expression (3.6-fold; *p* = 0.0098 vs. CTL and 4.4-fold; *p* = 0.0044 vs. acrylamide) (Figure 6a). In contrast, we did not observe any significant (*p* > 0.05) effect of acrylamide on *CXCL8* gene expression (Figure 6a).

Furthermore, a 6 h incubation with either acrylamide or CML was not associated with a greater level of IL-8 secretion (*p* > 0.05) in both the apical and the basal media (Figure 6b,c, respectively).

In contrast, the 6 h incubation with CML was followed by a statistically significant upregulation of *MAPK8* mRNA expression (5.9-fold; *p* = 0.0054 vs. CTL and 6.0-fold *p* = 0.0060 vs. acrylamide) (Figure 6d) and of *CASP3* mRNA expression (3.5-fold; *p* = 0.0121 vs. CTL) (Figure 6d).

## 4. Discussion

In this study, we sought to improve an in vitro model of the human intestinal epithelium that could mimic the physiological response to different food contaminant treatments. The Caco-2 cell line has been, for a long time, used for in vitro studies, and is known to differentiate spontaneously, express enterocyte phenotypes [13], and mimic the functional gut epithelium [14]. However, this cell line tends to grow heterogeneously and evolve differently, leading to variable tight junction protein expression, cellular localization [15], and intestinal permeability [16,17]. Several isolated clones reduce this heterogeneity [18], and among them, the TC7 clone is able to better reproduce the functionality of intestinal cells in vivo [19]. This Caco-2 clone is a good model for gut barrier and intestinal absorption [20], but also for intestinal drug metabolism [21]. Nevertheless, in vivo, intestinal epithelial cells are closely juxtaposed to other cells, such as goblet cells. As such, to get closer to a more physiological model of the intestinal epithelium, we chose to complete it by using a second cell line, HT29-MTX, expressing a goblet cell phenotype, to wrap the epithelial lining in the mucus component [22,23]. The co-culture of Caco-2 and HT29-MTX has already been validated in the literature as a representative model of intestinal epithelium [18,19,29,30,31]. To our knowledge, we are among the first to use the co-culture model of the Caco-2/TC7 clone with HT29-MTX to limit the bias described above with the parental Caco-2 cell line [20].

One of the main challenges was also to reproduce the physiological features of both the cell types. However, in the literature, ratios of Caco-2 and HT29-MTX vary widely: 75/25 [32]; 90/10; 80/20 and 70/30 [1]; 90/10, 75/25, 50/50, 25/75 [2]. We then evaluated in a preliminary study the responses of different ratios (60/40, 70/30, 80/20, 90/10) of these two cell lines grown together to determine the most appropriate one. From this study, we kept 90% of the Caco-2/TC7 cells and 10% of the HT29-MTX cells, which is the most widely used ratio with Caco-2 cells [33] and the closest to that of the intestinal epithelium in vivo. Indeed, under in vivo conditions, the intestinal mucosa epithelium is made of 85% to 95% enterocytes and 5% to 15% goblet cells, depending on the segment of intestine considered (increasing caudally from duodenum to distal colon) [34]. Furthermore, to ascertain that our cell model was still sensitive to an inflammatory signal, we also compared the sensitivity of the Caco2/TC7 mono-culture and that of the Caco2/TC7 and HT29-MTX co-culture to a pro-inflammatory stimulus, generated by a cocktail of 3 cytokines (TNF-α, IL1-β, and IFN-ϒ) following exposures of 2, 6 or 24 h [26]. Indeed, Caco2 cells, including the TC7 clone, are known to be able to produce inflammatory markers in response to pro-inflammatory stimuli [35]. We confirmed that the inflammatory effect of the cocktail was observed on the mono-culture as early as 2 h, as indicated by *NF-κB* gene activation, IL-8 activation, and secretion, and this lasted up to 24 h, although a decrease in the effect was observed between 6 and 24 h stimulation. Since the highest inflammatory response was observed between 2 and 6 h, we decided to expose the cells for 6 h for the rest of the study. Pro-inflammatory cytokine stimulation of the co-culture Caco-2/TC7 and HT29-MTX for 6 h also resulted in an activation of *CXCL8* gene expression and secretion following NF-κB activation, which is corroborated by the literature [36]. Furthermore, on both the mono- and the co-culture, this seems to be essentially associated to the activation of the NF-κB pathway, since neither the *STAT3* nor the MAP kinase gene expressions were modified after cocktail stimulation. At last, we did not observe any modification of the *CASP3* and *CASP9* pro-apoptotic genes after the stimulation by pro-inflammatory cytokines. Although all results were significant, the response of the co-culture was lesser than that of the mono-culture. Such a difference in the amplitude of response could be linked to the presence of a mucus layer in the co-culture.

Since the Caco2/TC7 and HT29-MTX co-culture responded to a pro-inflammatory stimulus, we investigated if this model could be used to evaluate the response to food contaminants. Among the large diversity of contaminants, we focused on residuals of environmental contamination and those generated during food processing, as we are exposed to several contaminants at once. Among pesticides, we chose to study the effects of CPF, an organophosphorus compound, known to generate a pro-inflammatory response in Caco2 cells [37]. Furthermore, the heat processing of food generates multiple molecules, among which are advanced glycation end products (AGEs), intermediate compounds of the Maillard reaction, whose effects on gut function and more generally on health are questionable. More specifically, we studied the consequences of an in vitro exposure to N^ε^-Carboxymethyllysine (CML), whose effects on the gut are controversial. Based on these considerations, we thus exposed the co-culture to CPF and to CML separately and together, in order to mimic food-bound exposure. The exposure of the co-culture to CPF (300 µM) was not associated with cell death. This dose was consistent with the dose of CPF (250 µM) described to be representative of oral daily exposure in humans, as used in a previous study [7]. However, the concentration of CPF used in this study was slightly increased since, in their work, the authors estimated that 20% of CPF was adsorbed by the plastic material, and in our study we used mucus-secreting cells. After a 6 h stimulation with CPF, the paracellular permeability of the co-culture was not altered. We, however, observed that CPF increased the mRNA expression of *TJP1* and *VIL1* without changing the mRNA expression of either mucins or the genes involved in the pro-inflammatory and pro-apoptotic signaling pathways. In a previous work using Caco2 submitted to CPF, authors described an increase in intestinal permeability (decreased TEER and increased 3H-mannitol passage after 6–8 h of treatment) [7], and we also showed that CPF daily exposure alters intestinal permeability in vivo. Such a discrepancy between the studies may mainly come from a reinforcement of the intestinal barrier function by the use, in the present work, of goblet cells, which generate, via mucus secretion, a physical layer lowering the ability of CPF to reach the epithelial cells. Another possible explanation could be the fact that a single exposure to the pesticide may not be sufficient to alter the epithelial intestinal lining, since we were not able to confirm previous in vivo results. In that work, we showed an increased intestinal permeability associated with the modification of the expression of genes coding for TJ proteins and for mucus composition [5,6], probably because the animals were submitted to a repeated low-level exposure to the pesticide, which is not the case here. Moreover, we did not find any induction of the pro-inflammatory or pro-apoptotic signaling pathways in the co-culture model, which is contradictory to the literature. Although most in vitro studies based on exposure to CPF mainly aimed at evidencing its cytotoxicity, very few focused on its effects on intracellular signaling, especially at the digestive level. Indeed, CPF activates ERK1/2 and p38 MAP kinases and slightly stimulates JNK activity in primary cultures of rat cortical neurons [38]. Furthermore, CPF, through the inhibition of MAPK pathways, is able to stimulate the assembly of TJ proteins (Occludin, ZO-1) and their recruitment to the cell membrane [39] in MDCK cells. One of the mechanisms proposed, based on human corneal epithelial cells, is the ability of ERK1/2 to increase paracellular permeability and induce tight junction disruption, characterized by an altered distribution of ZO-1 and Occludin [40]. Based on this observation, we may then suggest that the stimulation of TJ gene expression and mainly *TJP1* coding for ZO-1 protein is linked to the absence of MAPK and NF-κB pro-inflammatory signaling pathway activation in the co-culture model. Concerning the effect of the pesticide on the pro-apoptotic pathways, it was evidenced that the exposure (4–6 h) of human monocyte like cell line (U937) to CPF (71–284 µM) resulted in a dose-dependent increase in caspase-3activity [41], and that exposure to increasing doses of CPF (0–800 µM) for 24 h induced a dose–response activation of caspase-3/caspase-9 activities on a QSG7701 human hepatocytes cell line [42]. While we worked with similar concentrations of CPF, we did not observe this effect. Such differences may be due to the fact that, on the one hand, the intestinal co-culture model is less sensitive to contaminants than immune and hepatic cell lines and, on the other hand, the apoptotic pathways may not already be stimulated after a 6 h stimulation.

Since the consequences of CML for health are debated, we exposed the co-culture to this AGE, considered to be a glycotoxin, and compared its response to the one observed with the most widely studied AGE, acrylamide, classified as a putative carcinogen by the International Agency for Research on Cancer (IARC) [43]. N^ε^(6)-Carboxymethyllysine (CML) is suspected to stimulate pro-inflammatory signaling pathways in epithelial cells after binding to the receptor for advanced glycation end products (RAGE), a transmembrane receptor of the immunoglobulin superfamily [44]. Neither CML (300 µM) nor acrylamide (5 mM) induced any cell death in the co-culture. The concentrations used in this work are representative of a daily exposure to the two glycotoxins. Our results are contradictory to previous data evidencing a cytotoxic activity of acrylamide at 2.5 mM [44] and 5 mM [24,44], and of CML at 500 µM [27], on Caco2 cells. We may suggest that the presence of mucus-secreting cells may mitigate the deleterious effect of AGEs on cell viability, thus lowering the inflammatory response of the Caco2/TC7 cell line when in co-culture with HT29-MTX. However, while the mRNA expression of *MUC3* was higher after CML stimulation compared to acrylamide but not to controls, *MUC5AC* mRNA expression was significantly reduced by both CML and acrylamide exposure as compared to controls. Since MUC5AC is the major gel-forming mucin in the co-culture model [45], we suggest that mucus composition may be slightly modified by CML, and to a lesser extent by acrylamide exposure, without altering its protective effect. Furthermore, neither acrylamide nor CML increased intestinal permeability evaluated by FITC-Dextran efflux, although we observed a lower expression of *TJP1* and *VIL1* mRNA after CML stimulation compared to acrylamide exposure, but not compared to the non-stimulated cells. These effects may be linked to the absence of NF-κB pathway activation, since no modification in the gene expression of the transcription factor known to downregulate TJ genes expression in epithelial cells [46] was observed in our experiments. However, since we derived a higher expression of the gene coding for *CXCL8* following CML stimulation, there may still be an as yet undetermined inflammatory pathway activated in the intestinal cells. We also observed an overexpression of genes coding for MEKK1 (*MAP3K1*), JNK1 (*MAPK8*) et p38-α (*MAPK14*), but also for the chemokine CXCL8, whose expression is known to be activated by the JNK and p38-α pathways, after stimulation of the co-culture by CML (300 µM) for 6 h [47]. It is recognized that p38, ERK, and JNK are the three main classes of phosphorylases governing intracellular pro-mitotic and pro-inflammatory pathways [48]. Taking this into account, we can propose that, following 6 h stimulation by CML, epithelial cells may, by a yet unknown mechanism, stimulate the MAPK and not the NF-κB signaling pathways, thus activating chemokine gene expression. However, this phenomenon does not seem to last long enough to cause any secretion of IL-8, since at this time point we observed no upregulation of its secretion. At last, this set of experiments confirmed that acrylamide and CML do not act in the same way on the intestinal epithelium, since neither the NF-κB nor the MAPK pathways were activated by acrylamide. This result was unexpected since, according to the literature, acrylamide (1 and 3 mM) is able to upregulate the MEK/ERK signaling pathways via ERK phosphorylation, and the activation of two major transcription factors AP-1 and NF-κB at the origin of its carcinogenicity [49]. However, we cannot exclude that acrylamide may have affected Ca signaling, as already described in different types of cells [50], but this would be a hypothesis to investigate if focusing on the consequences of acrylamide. The induction of apoptosis pathways following the exposure to CML by comparison to acrylamide was also evaluated on the Caco2/TC7 and HT29-MTX co-culture by investigating the *CASP3* and *CASP9* genes’ expressions. The 6 h stimulation of the epithelial model tended to upregulate *CASP3* gene expression in the presence of acrylamide (5 mM), and resulted in an overexpression of the *CASP3* gene in the presence of CML (300 µM). As observed on the pro-inflammatory pathways, the pro-apoptotic pathways do not seem to be activated in a similar fashion. It was previously shown that acrylamide (5–20 mM) induced apoptosis and cell death in Caco2 cells after 8–16 h exposure [24]. Furthermore, the 24 to 48 h stimulation of RAW 264.7 cells by CML (0–100 mM) was associated with an activation of caspase-3 and caspase-9 proteins [51]. Such differences from the present work may be due to the establishment of a more robust barrier by the association of the two cell lines. Moreover, since only *CASP3* gene expression seems to have been stimulated, while neither *CASP9* gene expression nor the ERK pathways seem to be responsible for this activation, we may hypothesize that the upregulation of *CASP3* is caused by an as yet undetermined pathway.

The ultimate aim of the study was to evaluate the consequences of the concomitant administration of the two food contaminants for the co-culture model. The co-exposure of the co-culture to CPF (300 µM) and to CML (300 µM) did not result in an exacerbation of the pro-inflammatory signals caused by each of them separately. By contrast, the effects observed under CML were inhibited when co-stimulation with CPF was realized. This result was unexpected, since we were expecting effects at least equivalent to those observed with CML alone. We may thus speculate that the cellular effects of CPF and CML may occur via antagonistic pathways. These pathways still need to be clarified. One interesting lesson of the study is that a cocktail of toxicants induces a different response than toxicants administered separately, and the complexity of in vitro models will condition the sensitivity of the response (Figure 7).

## 5. Conclusions

In conclusion, the use of a co-culture of Caco2/TC7 with HT29-MTX is a suitable model for evaluating the pro-inflammatory response to a cocktail of food contaminants. One of the main interests of using such a clone, compared to the parental cell line, is its better stability after several passages, but also the better homogeneity of the response. A 6 h exposure to AGEs, such as CML, decreases mucins’ gene expression, but does not affect paracellular intestinal permeability. CML exposure seems to induce pro-inflammatory signaling pathways. CPF exposure seems to only affect tight junction gene expression, without altering paracellular intestinal permeability or inducing any inflammatory response. The use of such a model seems to be a promising alternative for evaluating the effects of food contaminants on the digestive tract. Thus far, there are still a limited number of studies that describe the effects of food contaminants on in vitro models of intestinal mucosa. This is mainly due to their lack of similarity to the in vivo response of the gut barrier. To overcome this point, it will be necessary to consider the immune component, as well as the microbiota’s interaction with the epithelium, in order to determine the physiological interactions governing intestinal homeostasis. Such models will be of great utility in understanding these events, particularly during very sensitive periods of life, and evaluating their consequences for the development of many chronic diseases of intestinal origin, such as non-communicable chronic diseases.

## Figures and Tables

**Figure 1 toxics-09-00135-f001:**
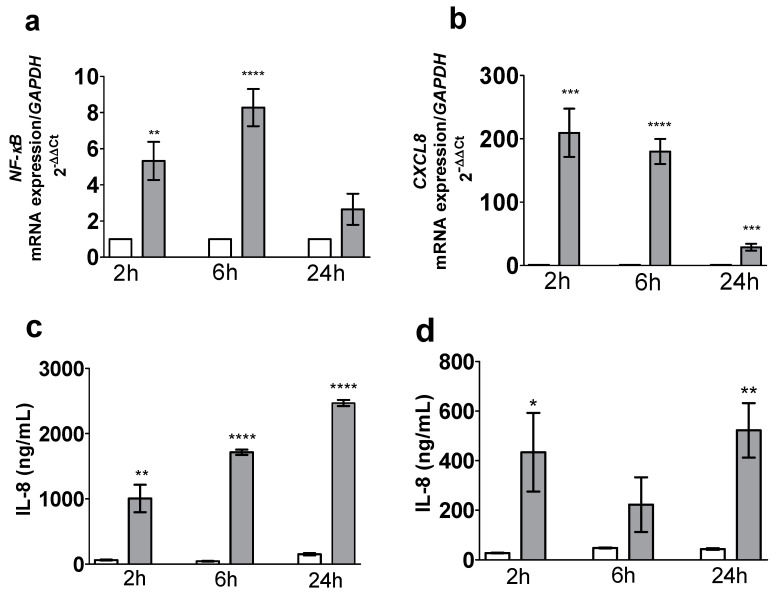
Kinetics (2 h, 6 h or 24 h of incubation with the cocktail of pro-inflammatory cytokines—CK) of expression of *NF-κB* mRNA (**a**) and of *CXCL8* mRNA (**b**) measured by qPCR, and of IL-8 secretion measured by ELISA in the apical (**c**) and basal (**d**) media from the Caco-2/TC7 mono-culture. Effect of a 6 h incubation with the cocktail of pro-inflammatory cytokines on *NF-κB* (**e**) and *CXCL8* (**f**) relative mRNA expressions measured by qPCR, and on IL-8 secretion measured by ELISA in the apical (**g**) and basal (**h**) media from the Caco-2/TC7 and HT29-MTX co-culture. Data are expressed as the mean ± SEM (*n* = 6). mRNA levels are expressed as fold induction over the CTL group set at 1. *, **, ***, ****: statistically significant difference vs. control (CTL) (*p* < 0.05; *p* < 0.01; *p* < 0.001 and *p* < 0.0001, respectively). CK, cytokines; CTL, control; CXCL8, Interleukin 8; NF-κB, nuclear factor-kappa B.

**Figure 2 toxics-09-00135-f002:**
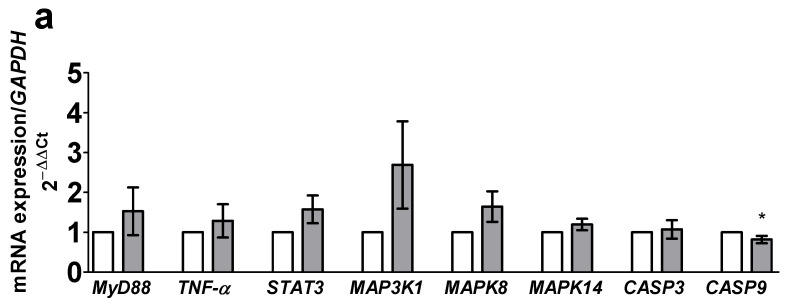
Effect of a 6 h incubation with the cocktail of cytokines on the relative mRNA expression of genes coding for the pro-inflammatory and pro-apoptotic signaling pathways measured by qPCR from the Caco-2/TC7 mono-culture (**a**) and from the Caco-2/TC7 and HT29-MTX co-culture (**b**). Data are expressed as the mean ± SEM (*n* = 12). mRNA levels are expressed as fold induction over the CTL group set at 1. *, **** statistically significant difference vs. CTL (*p* < 0.05; *p* < 0.0001, respectively). CK, cytokines; CTL, control; *CASP3*, Caspase 3; *CASP9*, Caspase 9; *MAP3K1*, mitogen-activated protein kinase kinase kinase 1; *MAPK8*, mitogen-activated protein kinase 8; *MAPK14*, mitogen-activated protein kinase 14, also known as p38-α; *MyD88*, myeloid differentiation primary response gene 88; *STAT3*, signal transducer and activator of transcription 3; *TNF-α*, tumor necrosis factor α.

**Figure 3 toxics-09-00135-f003:**
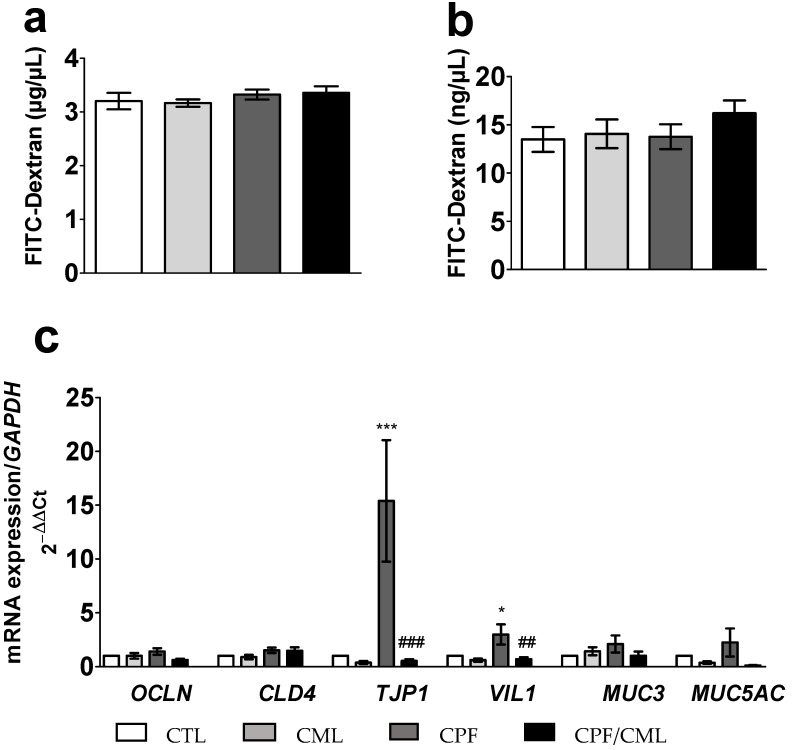
Effect of a 6 h incubation with CML (300 µM) and/or CPF (300 µM) on the level of FITC-Dextran in apical (**a**) and basal (**b**) media, and of the relative mRNA expression of genes coding for tight junctions and mucins measured by qPCR from the Caco-2/TC7 and HT29-MTX co-culture (**c**). Data are expressed as the mean ± SEM (*n* = 12). The level of gene expression was expressed as the fold induction over the CTL group set at 1. *, *** statistically significant difference (*p* < 0.05; *p* < 0.001 CPF vs. CTL); ##, ### statistically significant difference (*p* < 0.01; *p* < 0.001 CPF/CML vs. CPF). CML, N^ε^-Carboxymethyllysine; CPF, Chlorpyrifos; CTL, control; *CLDN4*, Claudin 4; *MUC3*, Mucin 3; *MUC5AC*, Mucin 5AC; *OCLN*, Occludin; *TJP1*, tight junction protein 1; *VIL1*, Villin 1.

**Figure 4 toxics-09-00135-f004:**
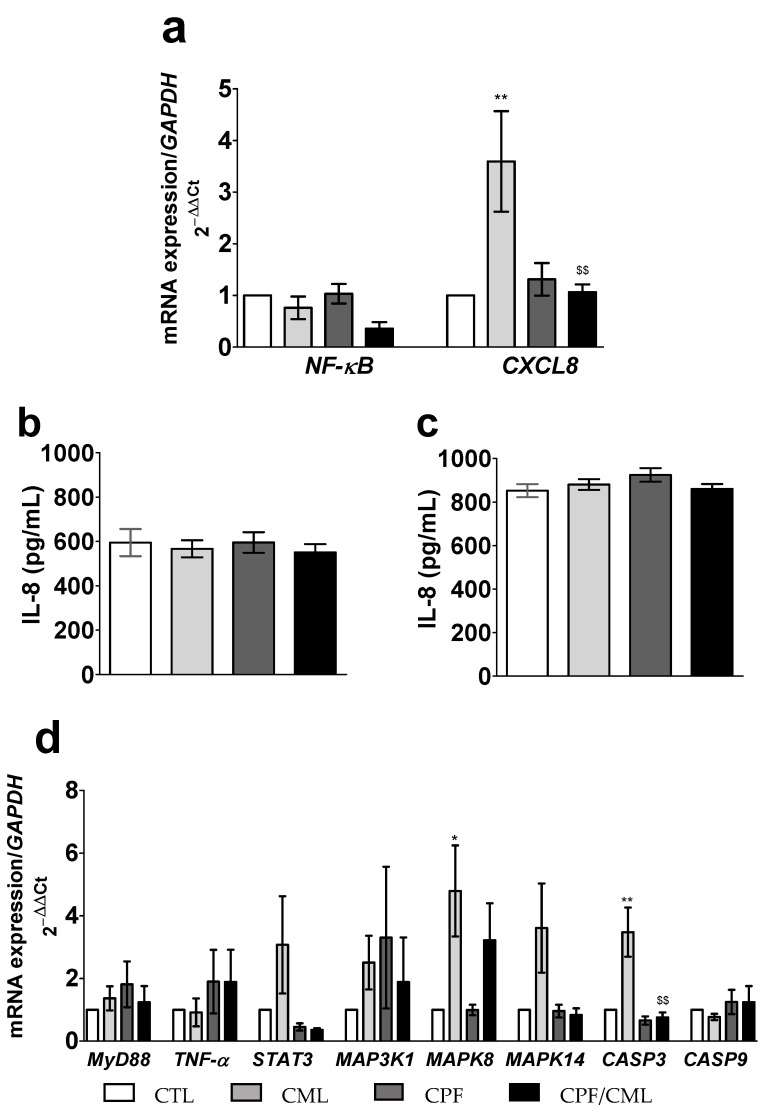
Effect of a 6 h incubation with CML (300 µM) and/or CPF (300 µM) on the relative mRNA expression of genes coding for *NF-κB* and *CXCL8* (**a**) measured by qPCR, and of IL-8 secretion measured by ELISA, in both the apical (**b**) and basal (**c**) media. Relative mRNA expression of genes involved in the pro-inflammatory and pro-apoptotic signaling pathways measured by qPCR from the Caco-2/TC7 & HT29-MTX co-culture (**d**). Data are expressed as the mean ± SEM (*n* = 12). The level of gene expression is expressed as the fold induction over the CTL group set at 1. *, ** statistically significant difference (*p* < 0.05; *p* < 0.01 CML vs. CTL); ^$$^ statistically significant difference (*p* < 0.01 CML vs. CPF/CML). CML, N (6)-Carboxymethyllysine; CPF, Chlorpyrifos; CTL, control; *CASP3*, Caspase 3; *CASP9*, Caspase 9; *CXCL8*, Interleukin 8; *MAP3K1*, mitogen-activated protein kinase 1; *MAPK8*, mitogen-activated protein kinase 8; *MAPK14*, mitogen-activated protein kinase 14, also called p38-α; *MyD88*, myeloid differentiation primary response gene 88; *NF-κB*; nuclear factor-kappa B; *STAT3*, signal transducer and activator of transcription 3; *TNF-α*, tumor necrosis factor α.

**Figure 5 toxics-09-00135-f005:**
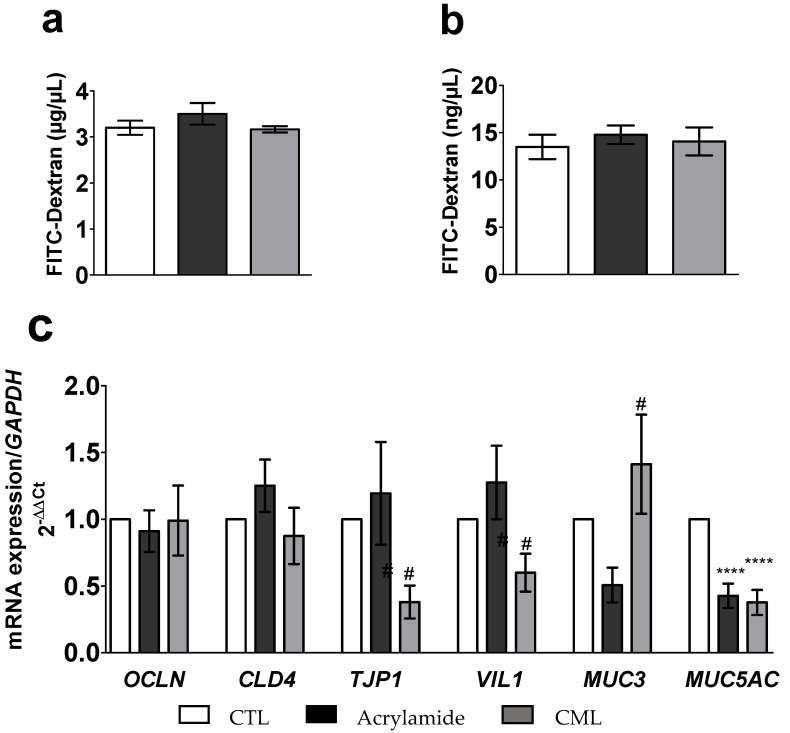
Effect of a 6 h incubation with CML (300 µM) or acrylamide (5 mM) on the levels of FITC-Dextran in the apical (**a**) and the basal (**b**) media, and the mRNA expression of genes coding for tight junctions and mucins (**c**) from the Caco-2/TC7 and HT29-MTX co-culture. Data are expressed as the mean ± SEM (*n* = 12). The level of gene expression was expressed as the fold induction over the CTL group set at 1. **** statistically significant difference (*p* < 0.001 vs. CTL), # statistically significant difference (*p* < 0.05 vs. acrylamide). CML, N^ε^-Carboxymethyllysine; CTL, control; *CLDN4*, Claudin 4; *MUC3*, Mucin 3; *MUC5AC*, Mucin 5AC; *OCLN*, Occludin; *TJP1*, tight junction protein 1; *VIL1*, Villin 1.

**Figure 6 toxics-09-00135-f006:**
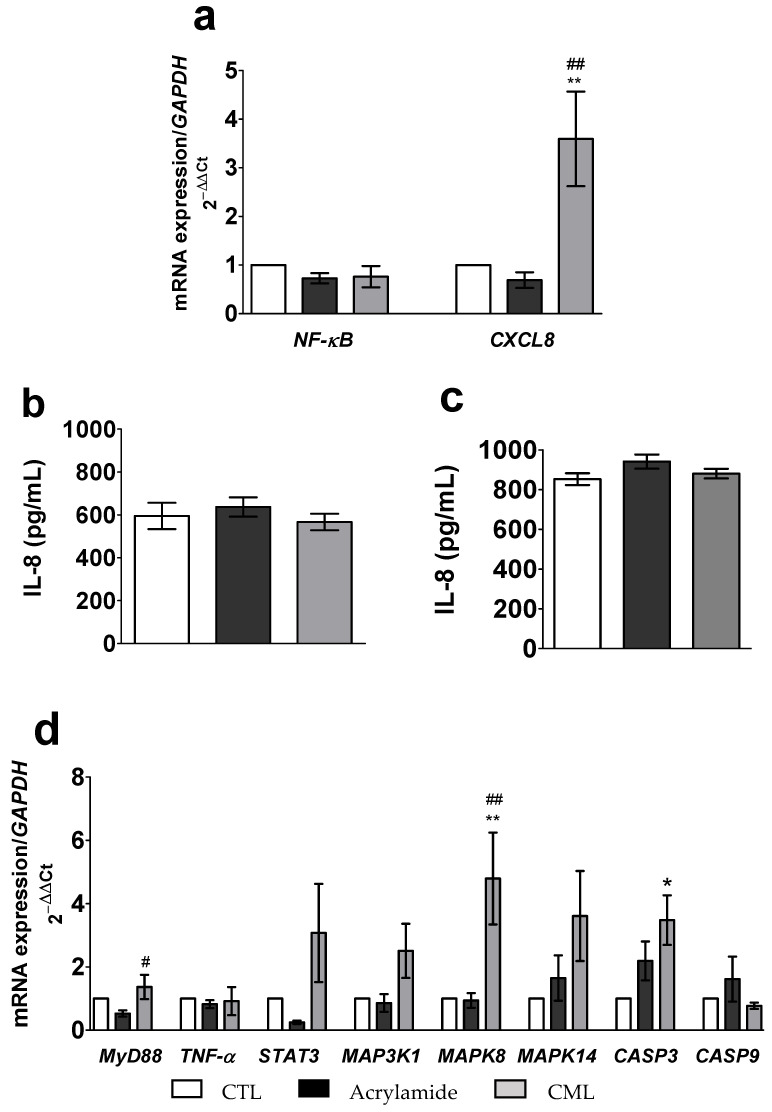
Effect of a 6 h incubation with CML (300 µM) or acrylamide (5 mM) on relative mRNA expressions of genes coding for *NF-κB* and *CXCL8* (**a**) measured by qPCR, and on IL-8 secretion measured by ELISA, in the apical (**b**) and the basal (**c**) media, and on the relative mRNA expression of genes coding for the pro-inflammatory and pro-apoptotic signaling pathways measured by qPCR (**d**) from the Caco-2/TC7 and HT29-MTX co-culture. Data were expressed as mean ± SEM (*n* = 12). The level of gene expression was expressed as the fold induction over the CTL group set at 1. *, **, statistically significant difference (*p* < 0.05; *p* < 0.01 vs. CTL); #, ## statistically significant difference (*p* < 0.05; *p* < 0.01 vs. acrylamide). CML, N^ε^-Carboxymethyllysine; CTL, control *CASP3*, Caspase 3; *CASP9*, Caspase 9; *CXCL8*, Interleukin 8; *MAP3K1*, mitogen-activated protein kinase 1; *MAPK8*, mitogen-activated protein kinase 8; *MAPK14*, mitogen-activated protein kinase 14, also called p38-α; *MyD88*, myeloid differentiation primary response gene 88; *NF-κB*, nuclear factor-kappa B; *STAT3*, signal transducer and activator of transcription 3; *TNF-α*, tumor necrosis factors α.

**Figure 7 toxics-09-00135-f007:**
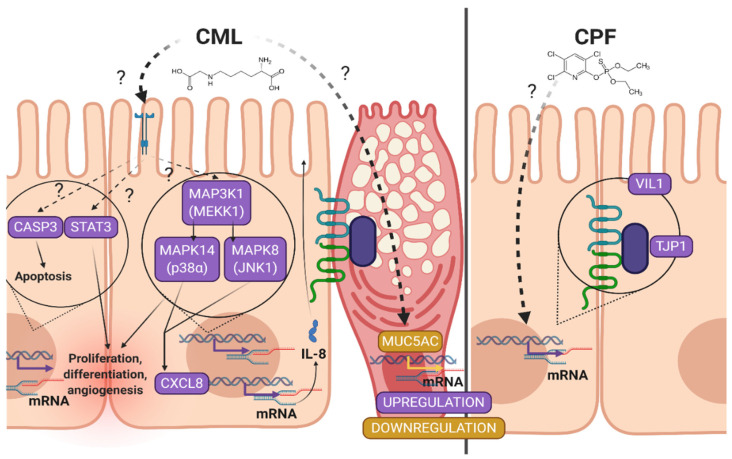
Potential effects of food contaminants on the co-culture of Caco-2/TC7 and HT29-MTX after a 6 h stimulation. *CASP3*, Caspase 3; *CASP9*, Caspase 9; *CLDN4*, Claudin 4; CML, N(6)-Carboxymethyllysine; CPF, Chlorpyrifos; CTL, control; *CXCL8*, Interleukin 8; *MAP3K1*, mitogen-activated protein kinase 1; *MAPK8*, mitogen-activated protein kinase 8; *MAPK14*, mitogen-activated protein kinase 14, also called p38-α; *MUC3*, Mucin 3; *MUC5AC*, Mucin 5AC; *MyD88*, myeloid differentiation primary response gene 88; *NF-κB*, nuclear factor-kappa B; *OCLN*, Occludin; *STAT3*, signal transducer and activator of transcription 3; *TJP1*, tight junction protein 1; *TNF-α*, tumor necrosis factor α; *VIL1*, Villin 1.

**Table 1 toxics-09-00135-t001:** List of primers used in the study.

Function	Gene	Name	Sequences 5′-> 3′ or Reference
Housekeeping Genes	*GAPDH*	Glyceraldehyde 3 phosphate dehydrogenase	GTCAACGCTGAGAACGGGAA AAA TGAGCCCCAGCCTTCTC (Eurofins)
	*PPIA*	Peptidyl-prolyl cis-trans isomerase A	CCTATCCTAGAGGTGGCGGA TCATCGCAGAAGGAACCAGAC (Eurofins)
Inflammatory Genes	*CXCL8*	C-X-C motif chemokine ligand 8	AGAGTGATTGAGAGTGGACC ACTTCTCCACAACCCTCTG (Eurofins)
	*MAPK14*	Mitogen-activated protein kinase 14, also called p38-α	QT00079345 (QIAGEN)
	*MAPK8*	Mitogen-activated protein kinase 8	QT00091056 (QIAGEN)
	*MAP3K1*	Mitogen-activated protein kinase-kinase-kinase 1	QT00088998 (QIAGEN)
	*MyD88*	Myeloid 88	QT00203490 (QIAGEN)
	*NF-κB*	nuclear factor-kappa B	GGGGGCATCAAACCTGAAGA GGAGAGAAGTCCCCAAAGGC (Eurofins)
	*STAT3*	Signal transducer and activator of transcription 3	QT00068754 (QIAGEN)
	*TNF-α*	tumor necrosis factors α	QT00029162 (QIAGEN)
Apoptosis Genes	*CASP3*	Caspase 3	QT00997997 (QIAGEN)
	*CASP9*	Caspase 9	QT00036267(QIAGEN)
Microvillly Gene	*VIL1*	Villin 1	AAGAAAGCCAATGAGCAGGAGAAG TTCTCAATGCGCCACACCTG (Eurofins)
Tight Junctions Genes	*CLDN4*	Claudin 4	GGCGTGGTGTTCCTGTTG AGCGGATTGTAGAAGTCTTGG (Eurofins)
	*TJP1*	Tight Junction Protein 1	GAATGATGGTTGGTATGGTGCG TCAGAAGTGTGTCTACTGTCCG (Eurofins)
	*OCLN*	Occludin	QT00081844 (QIAGEN)
Mucins Genes	*MUC3*	Mucin 3	QT02309335 (QIAGEN)
	*MUC5AC*	Mucin 5AC	QT00088991 (QIAGEN)

## Data Availability

Data supporting the conclusions are presented in the manuscript. The datasets used and/or analyzed during the current study are available from the corresponding author on reasonable request.

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
