# Peer review of "Food Contaminants Effects on an In Vitro Model of Human Intestinal Epithelium"

_toxics, 2021, doi:10.3390/toxics9060135_

Round 1

Reviewer 1 Report

Review of the manuscript entitled: “Food contaminants effects on an in vitro model of human intestinal epithelium”. In my opinion the manuscript is very interesting, contains a lot of new information, but there are matters that need to be clarified.

Line 99: Why the HT29-MTX cells not cultivated in monoculture (only CACO-2)?

Line 128: Why were only casp-3 and -9 measured? It is well described that proinflammatory cytokines activate the external pathway of apoptosis with casp-8 involvement.

Line 161: Why such high concentrations of compounds are used (5 mM acrylamide, 300 μM of CML or 300 μΜ of CPF)? It is well known that these concentrations are highly toxic. Non-lethal concentrations should be used for study of mechanisms of action mentioned compounds.

Line 227-231: In the figures are different font sizes, everything should be prepared in the same way. This should be standardized across all figures throughout the manuscript.

Line 253: the reduction in casp-9 expression may be due to strong presence of ROS. Did the authors measure cell viability or ROS? How did the authors choose the concentrations for testing? do they have any justification with the concentrations found in the tissues? in the organism? in food? A justification of the concentrations used should be added to the manuscript.

Line 436: reference should by corrected (Turco et al. 2011).

Line 479: in all manuscript authors must check carefully, gene abbreviations are not synonymous with protein “CASP3 activity” means = gene of CASP3 activity genes have no activity. I think the authors meant “caspase-3 activity” that is, enzyme / protein activity. The entire manuscript and all proteins must be checked. Furthermore, we write genes in italic and proteins in normal. I don't know where are the genes and where are the proteins in the manuscript.

The discussion is generally good.

Line 581: There should be no references in the summary / conclusions.

Author Response

Dear Editor and reviewers,

We thank you for your extensive review of our manuscript entitled “Food contaminants effects on an in vitro model of human intestinal epithelium”. Please find below our response to your comments. The changes have been tracked in the revised version of the manuscript. We carefully read the manuscript to improve english editing.

Line 99: Why the HT29-MTX cells not cultivated in monoculture (only CACO-2)?

As described in the methods section, the HT29-MTX cells express a goblet cells phenotype. As thus, the use of a HT29-MTX monoculture is not representative of physiological conditions in which are mainly found at the epithelial level enterocytes; Indeed, because of their peculiar phenotype, these cells are not associated with molecules absorption and/or metabolism.

Line 128: Why were only casp-3 and -9 measured? It is well described that proinflammatory cytokines activate the external pathway of apoptosis with casp-8 involvement.

We only presented in the manuscript the modifications of CASP3 and CASP9 mRNA expression. As the reviewer mentions in his comment, CASP8 can be activated following pro-inflammatory cytokines stimulation. However, we did not observe any modification of CASP8 gene expression neither after stimulation by CPF not by CML. The use of the pro-inflammatory cytokines cocktail was just aimed at ascertaining the response of the cellular model. Considering the kinetics of CASP genes expression, we, however, cannot exclude the fact that CASP8 expression could have been upregulated at an earlier timepoint. However, since we evaluated the effects after 6H of exposure to the food contaminants, it would be too speculative to mention this in the discussion.

Line 161: Why such high concentrations of compounds are used (5 mM acrylamide, 300 μM of CML or 300 μΜ of CPF)? It is well known that these concentrations are highly toxic. Non-lethal concentrations should be used for study of mechanisms of action mentioned compounds.

The rationale of concentrations used on cell cultures are described in the discussion section (from line 442 to 447 for CPF and from line 491 to 494 for AGEs). The preliminary part of the study was dedicated at assessing the concentrations presented in the literature and get the lowest concentration as possible to induced a cell response without any cell death. Doses used in vitro cannot directly converted into a daily exposure in human and the mention of lethal concentrations does not really apply here. This has been detailed in the below comment.

Line 227-231: In the figures are different font sizes, everything should be prepared in the same way. This should be standardized across all figures throughout the manuscript.

We apologize for the inconvenience due to the importation of figures in the support file. All original figures have the same font and size. This has been improved in the revised version of the manuscript.

Line 253: the reduction in casp-9 expression may be due to strong presence of ROS. Did the authors measure cell viability or ROS? How did the authors choose the concentrations for testing? do they have any justification with the concentrations found in the tissues? in the organism? in food? A justification of the concentrations used should be added to the manuscript.

As pointed out by the reviewer, upregulation of CASP9 gene expression may be due to a strong presence of ROS. However, as we did not observe any modification of cell viability at any of the chosen concentrations, we excluded this explanation.

The use of a cell model is a simplified approach for mimicking tissue reaction to a stimulus, and here of the intestinal mucosa. As thus, exposure to these environmental factors and direct conversion of ADI (Acceptable Daily Intake) or of toxic concentrations in food will not make sense. This is why the doses used in this study were set up according to the literature and in order to induce cell response without altering their viability to simulate the situation happening to cell when exposed to food contaminants on a life long basis. This point has been clarified in the manuscript.

The concentrations used are the lowest for which we got a cellular response with no alteration of cell viability and in line with the literature as explained in the discussion from line 442 to 447 for CPF and from line 491 to 494 for AGEs.

Line 436: reference should by corrected (Turco et al. 2011).

According to the reviewer recommendation, the reference has been amended in the revised version of the manuscript.

Line 479: in all manuscript authors must check carefully, gene abbreviations are not synonymous with protein “CASP3 activity” means = gene of CASP3 activity genes have no activity. I think the authors meant “caspase-3 activity” that is, enzyme / protein activity. The entire manuscript and all proteins must be checked. Furthermore, we write genes in italic and proteins in normal. I don't know where are the genes and where are the proteins in the manuscript.

The overall manuscript has been carefully checked in order to make sure that there is no confusion between gene and protein names.

Line 581: There should be no references in the summary / conclusions.

The reference was removed from the conclusion.

Reviewer 2 Report

The present manuscript evaluated an in vitro model of human intestinal epithelium that could mimic the response to different food contaminants. This in vitro model is a co-culture of Caco-2/TC7 and HT29-MTX.

The manuscript is clear and well written. The methods used for cell culture conditions, set up of an in vitro model and expression of genes coding for proteins of inflammatory and apoptotic pathways are appropriate. When assessing gut barrier integrity, the authors measured paracellular permeability and expression of certain genes coding for proteins involved in the barrier integrity. Despite that, measurement of transepithelial electrical resistance (TEER) -paracellular ion permeability- and a different set of tight junction’s proteins are particularly welcome. Thus, the claudin proteins are considered to be the structural backbone of TJ claudin-1, -3, -4, -5, and -8 tighten TJ are known to decrease  paracellular permeability, whereas claudin-2 forms charge-selective paracellular pores.

  • Besides that, in the text is stated that this in vitro model has been set up for the first time. This statement is quite inaccurate (see the work of Mitsou et al., Colloids Surf B Biointerfaces. 2019 184:110482) Furthermore this in vitro model has already been improved with the incorporation of THP-1 cells (immune cells). This should be clarified in the manuscript.
  • When measuring the cytotoxicity I will expect the IC50 values of the food contaminants; I also expected the combination of CPF/CML to be evaluated (as it was done or paracellular permeability further on).
  • Figures 3 and 5 should be joined as there are data represented in both.

Author Response

When assessing gut barrier integrity, the authors measured paracellular permeability and expression of certain genes coding for proteins involved in the barrier integrity. Despite that, measurement of transepithelial electrical resistance (TEER) -paracellular ion permeability- and a different set of tight junction’s proteins are particularly welcome. Thus, the claudin proteins are considered to be the structural backbone of TJ claudin-1, -3, -4, -5, and -8 tighten TJ are known to decrease paracellular permeability, whereas claudin-2 forms charge-selective paracellular pores.

As indicated by the reviewer, measure of alteration of paracellular permeability is frequently assessed via transepithelial electric resistance (TEER). This parameter was evaluated in the preliminary study. However, due to an absence of difference among the treatments applied to the cell culture, as illustrated by the quantified FITC dextran flux from the apical to the basal medium, we decided to present only the FITC parameter.

We do agree with the reviewer on the importance of claudins on the structure of TJ and their functionality. We did not present the results from CLDN2 gene expression since no change was observed. This is not surprising since, as the protein forms paracellular channels allowing transepithelial flux of ions and small solutes, the alteration of its expression would be related to some modification of TEER which was not observed here. At last, as we did not observe any gene expression modification of CLDN4 one of the most expressed claudin in the intestinal epithelium, we did not investigate other claudins gene expression, but rather focused onto intra-signaling pathways meant to be the most altered by the food contaminants studied.

Besides that, in the text is stated that this in vitro model has been set up for the first time. This statement is quite inaccurate (see the work of Mitsou et al., Colloids Surf B Biointerfaces. 2019 184:110482) Furthermore this in vitro model has already been improved with the incorporation of THP-1 cells (immune cells). This should be clarified in the manuscript.

We thank the reviewer for this careful review of the literature. Despite the extensive search of any paper related to the model, we were unable to get to the work of Mitsou et al. The reference has now been included in the manuscript and the text revised accordingly. We do not quite agree with the reviewer about the model used here. We meant to say that use of a more complex system by adding immune cells such as THP1 has never been done with the Caco2/TC7&HT29-MTX coculture although it was already described with the Caco2&HT29-MTX model. While the Caco2 cell line is the parental one of the Caco2/TC7 clone, the response of the cells does not seem to be strictly similar. This is why we mentioned that in the perspectives of the work.

When measuring the cytotoxicity I will expect the IC50 values of the food contaminants; I also expected the combination of CPF/CML to be evaluated (as it was done or paracellular permeability further on).

Figures 3 and 5 should be joined as there are data represented in both.

The reviewer is right. One can expect, when mentioning cytotoxicity, the IC50 values of food contaminants is expected. However, this was not the aim of the work described here. We were more interested in evaluating the consequences of non-cytotoxic but rather activating concentrations of food contaminants as an approach of a nontoxic daily exposure in human. As cell viability was not altered, this is the reason why we did not assess this parameter.

We agree with the reviewer that Figures 3 and 5 are quite similar since part of the groups are the same. However, merging the two sets of figures would mean to compare the Acrylamide (AA) effect to the CPF effect which as no scientific rationale since AA was here used to compare its effect to the CML one.